# Hang-Time HAR: A Benchmark Dataset for Basketball Activity Recognition Using Wrist-Worn Inertial Sensors

**DOI:** 10.3390/s23135879

**Published:** 2023-06-25

**Authors:** Alexander Hoelzemann, Julia Lee Romero, Marius Bock, Kristof Van Laerhoven, Qin Lv

**Affiliations:** 1Ubiquitous Computing, University of Siegen, 57076 Siegen, Germany; marius.bock@uni-siegen.de (M.B.); kvl@eti.uni-siegen.de (K.V.L.); 2Computer Science, University of Colorado Boulder, Boulder, CO 80302, USA; julia.romero@colorado.edu (J.L.R.); qin.lv@colorado.edu (Q.L.)

**Keywords:** wearable activity recognition, dataset, basketball, wrist-worn sensing

## Abstract

We present a benchmark dataset for evaluating physical human activity recognition methods from wrist-worn sensors, for the specific setting of basketball training, drills, and games. Basketball activities lend themselves well for measurement by wrist-worn inertial sensors, and systems that are able to detect such sport-relevant activities could be used in applications of game analysis, guided training, and personal physical activity tracking. The dataset was recorded from two teams in separate countries (USA and Germany) with a total of 24 players who wore an inertial sensor on their wrist, during both a repetitive basketball training session and a game. Particular features of this dataset include an inherent variance through cultural differences in game rules and styles as the data was recorded in two countries, as well as different sport skill levels since the participants were heterogeneous in terms of prior basketball experience. We illustrate the dataset’s features in several time-series analyses and report on a baseline classification performance study with two state-of-the-art deep learning architectures.

## 1. Introduction

Human activity recognition (HAR) systems aim to track people’s physical movements and categorize them according to predefined activity classes or clusters. Methods from machine learning, and especially deep learning, are applied in order to classify samples of sensor data into predefined classes. According to [1], only 30 datasets in total have ever been released publicly and 11 out of the 13 most cited datasets in the HAR community were released in 2015 or prior. Such datasets, especially the older ones, are commonly recorded in laboratories and follow strict activity protocols and movement patterns. Since scientists are lacking solid annotation methods and tools for recordings in the wild, they tend to fall back to a controlled lab environment, in which visual systems, often cameras, can be installed to facilitate labeling the sensor data in hindsight. Due to the labor-intensive work of labeling data, the number of participants is often limited. Significant hurdles for experiments conducted in the wild lead to an imbalance in the number of publicly available datasets from controlled environments in comparison to uncontrolled environments.

However, depending on the design of the experiment itself, a sports environment, e.g., Figure 1, can be seen as a semi-controlled environment, since its recording sessions can include both practice drills (controlled) and game sessions (uncontrolled). Due to the nature of the sports domain, this data contains highly variable and dynamical movement patterns, which exhibit high intraclass variability, as well as high intersubject variability [2] due to gender, height, weight, personal play style, and athletic ability of the subject. These differences are important in real-world scenarios, and classifiers in general perform worse on in-the-wild datasets than on lab-recorded datasets due to effects such as weak labeling [3] or smoothness in the performed activities [4]. Therefore, providing a publicly available dataset containing complex sports-related activities can have an important impact on how we design and validate our future HAR algorithms and gives researchers the security of a semi-controlled environment with precise labels based on video recordings.

Many previous studies, as summarized in Table 2 (for sports), Table 3 (for basketball), and earlier published surveys (e.g., [5,6]), demonstrate that there is an interest in using inertial measurement unit (IMU)-based wearable solutions for activity recognition in sports. Professional athletes use sensor-based training methods to improve their sporting skills. The German professional soccer clubs, Hannover 96 and 1. FC Magdeburg utilize the commercial body-worn IMU sensors, Vmaxpro [7], which monitors the athletes’ movements and presents training recommendations, including specific strength training exercises, via smartphone for the trainer and athlete. In 2021, the sports fashion company *Adidas* released a sensor-equipped inlay sole for a soccer shoe [8], which is capable of detecting soccer-specific activities. Similarly, in 2020 the Finnish-based company SIQ [9] released a sensor-equipped basketball with feedback aiming to improve players’ shooting skills.

**Contributions:** This dataset is the first publicly available dataset with sensor-based basketball activities collected from teams of players doing both structured practice drills and an unstructured game. The classes included were selected by researchers with many years of experience in playing basketball and represent a full range of basketball activities that cover key aspects of the sport. The activities included show high dynamics, complexity, and variability within the same subject (due to different execution styles) and also between subjects (due to experience and play style). Since recordings are split into warm-up, drill, and game sessions, the dataset provides a mix of controlled and uncontrolled environments. The game shows a higher dynamic because of the influence of other players and a higher pace than that in the drills. This setup can be seen as a transition from a controlled environment to a semi-controlled recording environment. Because the dataset does not contain information about successful scoring, it is not necessarily meant to be used for skill assessment. However, the metadata does contain information about the players’ experience (novice or expert). Novices are players with little prior experience in playing basketball. Players’ execution of activities can therefore be expected to display a large variance. Since the dataset was recorded from 24 participants (roughly the size of two complete teams) across two continents, it also includes the inherent differences within the rule sets played by the International Basketball Federation (FIBA) in Europe and the National Basketball Association (NBA) in Northern America. This is a unique setup that is not available in other sports.

**Impact:** This dataset can be used by the Ubiquitous Computing community to tackle a variety of research questions in the area of Human Activity Recognition. A (sports) dataset of this scope and study design is not yet publicly available, since it contains the same set of activities recorded with the same hardware and sensor modalities in both controlled and uncontrolled environments. The study is multi-part where some parts are controlled by prescribed activities and other parts are uncontrolled, such as a “free-movement” game. The nature of basketball is such that this mix of controlled conditions is easily captured in video and manually labeled in detail. The multi-part study adds complexity and diversity to the data and gives researchers a new playground to benchmark algorithms and approaches as well as (possibly) spot deficiencies in existing state-of-the-art architectures. Furthermore, the game phases include data that models team and game dynamics. This feature is something that can be explored deeper by future studies with regard to group activity recognition. This work provides layered labels, or multi-labels, as certain activities consist of a series of other activities. The complex characteristics of the data and the differences due to the location can help to address open research problems, such as Transfer Learning and Data Augmentation, recognizing complex activities in dynamic and real-world environments. Further development in these areas may include—but is not restricted to—research focused on data recording techniques and annotation procedures, data preprocessing (e.g., data segmentation), feature extraction or developing new deep learning methodologies and evaluation methods. Methodology-wise, we restricted our recording setup to commercial and mostly open-source recording components (in particular the smartwatches and their firmware). Such a low-effort recording setup has the significant advantage of being deployable in spontaneous situations and would not be restricted to basketball—if further developed. The labeling setup builds on previous work by the community and focuses on reproducibility by other research labs.

## 2. Motivation

Basketball is played across the globe, but the two of the most dominant rule sets are (1) Fédération Internationale de Basketball (FIBA) [10], which is played by: Basketball Champions League, Euroleague Women, Basketball Champions League Americas, FIBA Europe Cup, EuroCup Women, FIBA Asia Champions Cup, FIBA Intercontinental Cup, Olympic Games, etc. and (2)—the most important basketball league worldwide—the National Basketball Association (NBA) [11]—played in North America. The two sets of rules are similar but differ in several details [12]. For example, in contrast to the FIBA rules, NBA rules allow players to do a so-called 0-step—an additional step between catching the ball and the first dribble. Other differences include game time (40 min for FIBA games vs. 48 min for NBA games) and basketball court dimensions (28 m × 15 m for FIBA vs. 28.7 m × 15.2 m for NBA). In addition to these differences, the play styles in professional European and North American basketball tend to be slightly different as well. NBA teams often build their game around one or a few star players and a more aggressive defense. In contrast, European teams focus more on team play and a compact defense.

Basketball is a very dynamic and highly intense sport that combines fast movements, quick switching between offense and defense, and diverse execution of activities. Activities in this sport can be characterized into one of the following activity categories: (1) short actions or micro-activities (passing or rebounding), (2) complex activities (shooting the ball, layups), and (3) periodical activities (sitting, standing, walking, running, and dribbling). These activities are performed differently by every player, but also by the same player depending on factors such as in-game situations, physical fatigue and stress level, mental state, and improving skills over time. For that reason, the three research challenges defined in 2014 by Bulling et al. [13]—(1) intraclass variability, (2) interclass similarity, and (3) the NULL-class problem—are all reflected by the dataset presented here. These three challenges become more significant with less structured, real-world data, such as data from a real basketball game. In a game situation where external influences, such as other players, affect the gameplay and physical movements, these characteristics are more apparent.

A dataset such as the one presented in this article, which is recorded during two real basketball training sessions lasting between 1 and 2 h, also offers the opportunity to close the gap between controlled and uncontrolled study setups. After a few minutes in the warm-up session, participants reported that they forgot that they were monitored through their smartwatches and nearby cameras, and behaved like they would in a usual training session. We argue that the basketball game part of the dataset equally encouraged participants to move naturally [4,14]. Results show that even though science is advancing fast in the area of HAR, it is still challenging to train machine learning models that are capable of reliably detecting activities in naturalistic scenarios, such as [15]. In order to overcome this challenge, we consider the next important step in HAR to be that future algorithms are developed and evaluated on realistic datasets. Sports HAR datasets in general can be the perfect setting for researchers to do exactly this and they could allow for deeper insights for sports scientists as well as the deep learning and HAR community. Specific datasets that contain sports activities, e.g., DSADS [16] or the study presented by Trost et al. [17], often contain a variety of different sports in one dataset and reduce entire sports, such as playing basketball, to single target classes to be detected. The UTD Multimodal Human Action Dataset [18] contains four repetitions from eight subjects of 27 different activities from a variety of domains, such as sports. However, the included sports activities are limited to one specific activity per sport, e.g., shooting a basketball. Activities from these datasets are not representative of an entire sport. Inertial sensor-based and sports-specific datasets that capture the variability and complexity of a sport are not yet available on public repositories. Even recently published datasets, such as TNDA-HAR [19], focus on simple periodical locomotion activities, and additionally, available datasets that are used by the Ubiquitous and Pervasive Computing community rarely combine (1) scope, (2) quality, (3) variability, (4) complexity, and (5) reproducibility in the same benchmark dataset. The basketball data published by [16,17,18] does not represent the same level of complexity regarding the recently mentioned characteristics with the dataset we present since the class defined as *basketball* is highly simplified. We, therefore, highlight this as one of the main motivations for such a dataset. The following Table 1 gives an overview of relevant HAR datasets. As one can see in the Environment column, most of the datasets are recorded in controlled environments, partly because data collection is easier, and partly due to the lack of annotation methods without synchronized video recordings. Tools such as [20,21] or [22] are designed to be used in hindsight, with video footage and with well-defined synchronization gestures at the beginning and end of the video. Among the datasets presented in Table 1, only ActiveMiles [23] and Leisure Activities [15] are recorded in an uncontrolled environment. ActiveMiles is limited to simple locomotion activities, and Leisure Activities consist of six participants’ wrist-worn inertial data over a week where each of them performed one specific leisure activity daily. The WetLab dataset can be seen as recorded in a semi-controlled environment, where participants were told to follow a specific protocol for an experiment in the wet lab, but they were allowed to execute steps in their preferred order and at their own speed. This environment in combination with the sporadic activities makes it a difficult dataset to learn for machine or deep learning models with results of ~40% F1-Score.

Stoeve et al. [36] took IMU-based activity recognition from the lab to a real-world soccer scenario where passing and shooting in real soccer games are recognized.

This study did not publish the dataset publicly, however. We consider sports in general to be a highly interesting scenario for benchmark datasets that are aimed at further developing learning mechanisms that are also capable of detecting periodic activities, such as *sitting*, *standing*, *walking*, *running*, short or micro-activities such as *passing*, or *rebounding* and also complex activities such as *shooting* a basketball or performing a *layup*.

Finally, we summarize the main features of our Hang-Time HAR dataset as the following:(1)Hang-Time HAR consists of wrist-worn inertial data from 24 participants from two teams and from two countries with two different rule sets, performing 10 different basketball activities.(2)Hang-Time HAR is recorded in three different types of sessions: (1) warm-up, (2) drill, and (3) game. The drill sessions are executed in a structured way where participants were instructed to execute single specific activities, in a predefined order. However, the warm-up and game session followed the teams’ typical routine and are not tied to an activity protocol and participants were allowed to play as they preferred.(3)Hang-Time HAR includes considerable variety, with both simple and periodic activities, short or micro-activities, and complex activities. Hang-Time HAR also explicitly contains data from participants with different experience levels and following different basketball rule sets.(4)Hang-Time HAR is labeled on four different layers: (I) coarse, (II) basketball, (III) locomotion, and (IV) in/out. This will allow future researchers to combine labels, such as for example *dribbling + walking*, *dribbling + running*, or *jump shots*. This results in more complex activities and it becomes more challenging for the classifier to perform well.

## 3. Related Work on Sports Studies

IMU-based sports activity recognition is one of the main application fields for HAR studies, as summarized in Table 2 and Table 3. It has already been proven for a variety of different sports, such as running [37,38,39], ball sports [36,40,41,42,43,44], winter sports [45,46], sports for the disabled [47], or fitness [48,49,50,51], that activity recognition algorithms are capable of detecting specific activities tied to these sports based on IMU data as input. Other studies that incorporated IMUs focus rather on the athletes’ performance [52,53,54,55] or gait estimation [56,57] than activity recognition.

Basketball has been studied by several works using IMU sensor data beginning in 2014. The studies presented in Table 3 focused on a wide field of applications within basketball. Hoelzemann et al. [67] detected different dribbling styles and shooting with a single wrist-worn full IMU. Mangiarotti et al. [68] used two IMUs worn on each wrist to differentiate between passing, shooting, and dribbling the ball. Sviler et al. [69] focused on locomotion activities, such as jumping, acceleration, deceleration, and change of direction. Sangüesa et al. used IMU and RGB video data to detect complex basketball tactics. Lu et al. [70] and Liu et al. [71,72] attached smartphones to the body and used the built-in accelerometer to detect a variety of basketball activities. Lu et al. and Liu et al. showed that accelerometer data alone is sufficient to classify basketball activities. Perhaps the most comprehensive basketball activity study to date was conducted by Nguyen et al. [73]. The group used data from five full IMUs attached to the participants’ shoes, knees, and lower back in order to classify frequently occurring basketball activities such as walking, running, jogging, pivot, jumpshot, layup, sprinting, and jumping. The most relevant studies to this work are [67,71,72,73,74]. Ref. [67] focuses exclusively on the distinction of dribbling techniques and the recognition of the shooting motion, but it can be viewed as a feasibility study that inspired the development of a comprehensive basketball activity dataset. Table 3 shows that the only study that made their dataset publicly available is Trost et al. [17]. However, downloading the dataset from the source given in the manuscript is not possible at the time of writing this manuscript.

In comparison to our study, the majority of these studies predominantly utilize machine learning techniques or statistical analysis. The only exceptions are Eggert et al. [75] and Bo et al. [74], both of which employ deep learning-based approaches. However, these studies lack data with equivalent scope, complexity, and diversity. Moreover, they primarily focus on specific aspects, as mentioned earlier, and do not incorporate game phases, which frequently involve changing activity patterns, in their evaluation. sensors-23-05879-t003_Table 3Table 3IMU-based basketball activity recognition studies. Trost et al. is the only team that made their dataset publicly available for download. However, the dataset is currently unavailable for download. (* Not accessible from the source given by the manuscript at the time of writing).Sensor Based Basketball Studies**Study****(#) Activities Performed****Sensors/Systems Used****# Subjects****Dataset****Published****Analysis Method**Hoelzemann et al. [67](4) different dribbling
techniques, shootingWrist-Worn Full IMU3NoMachine Learning
(kNN, Random Forest)Sviler et al. [69](4) jumping, acceleration,
deceleration and
change of directionFull IMU13NoStatistical AnalysisNguyen et al. [73](8) walking, running,
jogging, pivot,
jumpshot, layupshot,
sprinting, jumpingFive Full IMUs3NoMachine Learning (SVM)Trost et al. [17](7) lying, sitting, standing,
walking, running,
basketball, dancingTwo Full IMUs,52Yes *Statistical Model
(Logistic Regression
Model)Bo [74](5) standing, running
standing dribble,
penalty shot,
jump shot5 IMUs (Acc. & Gyr.)20NoDeep Learning (RNN)Lu et al. [70](5) standing, bouncing ball,
passing ball, free throw,
moving with ball3 smartphones with accelerometer4NoMultiple Supervised
Machine Learning
ClassifierLiu et al., 2015 [71] and 2016 [72](8) walk, run, jump, stand
throw ball, pass ball,
bounce ball, raise hands2 smartphones with accelerometer10NoMultiple Supervised
Machine Learning
ClassifierSangüesa et al. [76](5) complex basketball tactics:
(pick and roll, floppy offense
press break, post up, fast break)IMUs and video footage11NoMachine Learning (SVM)Mangiarotti et al. [68](3) passing, shooting, dribblingtwo wrist-worn IMUs2NoMachine Learning
(SVM, kNN)Staunton et al. [77](1) jumpingMARG Sensor (magnetic, angular
rate and gravity).54NoStatistical AnalysisEggert et al. [75](1) jump shotfoot-worn IMU10NoDeep Learning (CNN)Bai et al. [78](1) basketball shotsone wristband-worn IMU,
one Android smartphone put in the
trouser pocket.2NoMultiple Supervised
Machine Learning
ClassifierHasagawa et al. [47](2) Wheelchair basketball:
push and stopwheelchair equipped with two IMUs6NoFeature and
Statistical Analysis


Observing a basketball game and interpreting activities executed on the court is a research topic primarily driven forward by computer vision studies. Therefore, according to Table 4 computer vision based activity recognition datasets are already publicly available to the community. The datasets presented in Table 4 mostly contain RGB data. The dataset used by Hauri et al. [79] is available for download and contains, among other modalities, 1D (y-axis) accelerometer data of NBA players shooting a basketball. However, the authors confirmed to us that the acceleration data in their dataset were not recorded with a wearable sensor device. Moreover, they were extrapolated from the video data by taking into account the positional data of the players and the time stamps. The study focused on detecting complex tactical group activities such as pick and rolls or handoffs. The studies conducted with visual data are more comprehensive with regards to the number of classes than the IMU-based activity studies. Gu et al. [80] classified 26 fine-grained basketball activities into 3 broad categories. A very early study, conducted in 2008 by De Vleeschouwer et al. [81], mixed basketball activities such as throwing, passing, or rebounding the ball, with detecting context-based activities such as player exchange, rule violations, or fouls. Maksai et al. [82] estimated the trajectory of a ball in different sports, including basketball. Ramanathan et al. [83] focused on scoring activities, such as performing layups, 3- and 2-point shots, free throws, and slamdunks.

Although a large number of activity studies explore sports data and specifically basketball data, the number of publicly available benchmark datasets is significantly low. A fine-grained IMU-based sports dataset that does represent one sport is not yet publicly available.

## 4. Methodology

This section provides detailed information about the study parameters, the hardware and software used to record the data, the preprocessing and labeling process, as well as recommendations for other researchers for recording IMU data. The second part of this section describes in detail the dataset in regard to the class characteristics.

### 4.1. Study Design

This dataset contains data collected during two separate periods and following the same study protocol. The first author supervised Study 1 at the University of Siegen, following FIBA regulations, which did not require IRB review. The second author conducted Study 2 at the University of Colorado Boulder, according to NBA regulations, and the study is IRB-approved. In subject recruitment, we excluded any person with a disability impairing their ability to play basketball and any person under the age of 18 years. Four modes of data were collected during the study: information collected manually by researchers, online questionnaire, smartwatch accelerometers, and video cameras in order to annotate the accelerometer data. However, the video data contains information that could de-anonymize our participants and is therefore not included in the dataset.

Prior to the study, participants signed a consent form that outlined the study protocol and risks of harm, and they were informed that the questionnaire and accelerometer data will lack any personally identifiable information and that a dataset containing these two modes of collected data will be made publicly available. At the start of the study, participants received one smartwatch and were assigned a unique identifier. The researchers manually collected the unique ID and name of the participants, in order to allow them to retroactively request for their data to be deleted prior to the release of the dataset. Participants filled out an online questionnaire collecting age, height, weight, gender, dominant hand, and history of playing basketball. Participants were then instructed to wear the smartwatch on their dominant hand and perform a sequence of basketball-related activities (i.e., standing, walking, running, dribbling, shooting, layups, and a game). Please see Appendix A Figure A1 for the study protocol. Two cameras were used to record each study, see Figure 2, and the footage was combined for the labeling process.

The study protocol is divided into two parts. The first part is designed to collect controlled data by having participants complete a sequence of predefined activities for a defined period of time, while this first part is controlled, it also simulates real-world basketball drills in practice sessions where players repeatedly practice a certain activity (e.g., layups, shooting, dribbling, running). The second part is a basketball game between two teams each with five players per team on the court, and extra players rotated into the game. Video cameras were set up along the sidelines of the court in order to record each participant’s activities for the labeling process. The differences between the two studies as well as the specifications of the recorded videos are documented in Table 5.

We have several recommendations for the collection of similar datasets based on our own experiences conducting the study and annotating the data. In the context of this study, we recommend setting up two wide-angle lens cameras, e.g., GoPro, side-by-side with each one capturing one half of the court and additionally instructing participants to wear uniquely colored clothing to aid the labeling process. We found that the cameras have a short battery life and it was necessary to bring extra batteries or a portable power bank to continuously charge the camera for the duration of the study. Finally, in order to synchronize the video footage with the smartwatch data, we recommend having participants complete a synchronization gesture, such as jumping, simultaneously on video at the start and end of the study.

**Hardware:** Each subject’s inertial data was captured by an open-source smartwatch, which was fitted to the user by the author conducting the study to fit comfortably around the dominant wrist. This watch was used to record 3D accelerometer data at ~50 Hz and at a sensitivity of ±8 g, This watch was used to record 3D accelerometer data at ~50 Hz and at a sensitivity of ±8 g, using the Bangle.js smartwatch with our custom firmware [90] The watch firmware was programmed to record the acceleration data and display the current time and date. It did not need pairing to other (e.g., Bluetooth) devices during the study. The axis orientation, viewed from above, is as follows: +X-axis points at a 90° angle to the left, +Y-axis points at a 90° angle forward and the +Z-axis points upwards at a 90° angle.

**Controlling the Bangle.js Smartwatches:** The Bangle.js smartwatches can be controlled with a custom smartphone app, which is implemented as an open-source cross-platform solution using Flutter [91] and is made available on the Apple AppStore, the Google Play Store, and on Github [92]. The app communicates via Bluetooth Low Energy with smartwatches. In order to download the data from the devices after stopping the recordings, the smartwatches can be connected via Web-BLE with a local PC. Through a website [93] the devices’ flash storage space can be accessed. The following Figure 3 depicts the procedure of starting the smartwatches. The first screen of the figure shows the app searching for nearby Bangle.js devices. After all nearby devices were found, 4 smartwatches in total, one can either start all devices individually or press the button “Start All” to start all visible devices simultaneously, screen (2). Both options open a dialogue, screen (3), where the researcher can choose the sampling rate (Hz), sensitivity (g), and starting time. Available sampling rates are 12.5, 25, 50, and 100 Hz and the sensitivity can be set to ±2, ±4, and ±8 g. The smartwatches need to be programmed to start at a preselected full hour. If the device should start immediately it needs to be set to the current hour. After pressing the Start button (3), the app connects to either one or all Bangle.js devices, synchronizes the time, and programs the preselected parameters. We did not evaluate how many Bangle.js smartwatches can be started simultaneously; however, we did not encounter any issues while starting up the 14 devices at the same time.

### 4.2. Obtaining Ground Truth

The raw accelerometer data is stored in CSV format. The labeling of ground truth was performed in hindsight with the multimedia annotation tool ELAN [94], which was originally developed as a linguistic annotation tool. The tool has the functionality to visualize additional time-series data [95] and display both modalities together. Before the annotation of the data, we ensured that sensor and video data are aligned with each other by using a jumping as a synchronization gesture with a few seconds of sedentary activity prior to and after the jump. The accelerometer data were then manually moved to the correct position. Figure 4 shows exemplary ground truth for Locomotion and the Basketball layer of subject 8 (with ID *05d8_eu*).

Most of the samples are labeled as *not_labeled*, especially on the basketball layer, since basketball activities tend to occur sporadically (whenever a player has the ball).

### 4.3. Dataset

The term Hang Time generally refers to the time a player spends in the air while shooting or passing a ball. This term, however, has been used by sports magazines [96], game developers [97] or producers of basketball equipment [98] as an inspiration to name their product. We decided to name our dataset Hang-Time HAR—which is focused on the time-series analysis of basketball activities—due to its high memorability, and its short and succinct form. The name represents to us the dataset’s direct relationship between basketball, **time**-series data classification and therefore human activity recognition. The name was consensually approved by the authors. Hang-Time HAR provides accelerometer data recorded with ~50 Hz and ±8 g. Even though a full IMU has not been used, the data provided can be specified as complex due to the given classes. Table 6 provides additional meta-information about every participant. The same information is available in the file *meta.txt* and downloadable from the dataset repository. In total, we recorded ~1:50:00 h of 13 participants from Germany and ~1:16:00 h of 11 participants from the USA.

The study was conducted in collaboration between two laboratories from the University of Siegen, Germany, and the University of Colorado Boulder, United States of America. In total 24 subjects participated in the study. Participants from Germany were mostly players from a semi-professional basketball team that participates actively in a basketball league. Participants from the USA were mostly graduate students with mixed prior experience in basketball. We originally included a void class for miscellaneous movements outside of the primary labeled ones, such as drinking from a water bottle or tying shoes. These were mostly performed during rest breaks. The samples annotated as void resulted in an irrelevant small class, which could not be recognized by our classifier because they are most often performed in conjunction with one of the locomotion classes. We ultimately decided against including this void class, since it was very rare that players were not performing one of the 10 classes of *locomotion* or *basketball* activities. However, the data that is not annotated as one of the aforementioned classes are categorized as *not_labeled*. This class can be seen as a very noisy but realistic void class that can be used by researchers whom focus on deeper insights in the NULL-class problem defined by Bulling et al. [13] or whom would like to evaluate deep learning architectures that are focused on the robust classification of void data. This class mostly contains data during resting periods or transitions between sessions. However, since the data is recorded under real-world conditions, many participants did not sit and rest during these periods, instead, they tended to walk through the gym, shoot the ball, or perform individual dribbling exercises. One of the players, namely *2dd9_na*, wore the smartwatch accidentally on their non-dominant hand, out of habit. We decided to keep this participant in the dataset since this participant represents something that could easily happen in real-world scenarios. Therefore, we think that the participant has added value to the dataset and can be useful for certain studies at a later date.

**Preprocessing:** We decided to keep the preprocessing on the raw data from the smartwatches to a minimum, as these were already provided with a timestamp and in the *g* unit. The smartwatch’s accelerometer samples’ timestamps contained slight (<2%) deviations, so we adjusted the time-series by resampling to ensure that all data maintains exact 50 Hz equidistant timestamps. Other common methods of preprocessing inertial data for activity recognition, such as rescaling or normalization to improve machine learning results, were not applied.

**Labeling:** The data was labeled by two experts, one from each institute, and labeled on 4 different layers: (I) *coarse*, (II) *basketball*, (III) *locomotion*, and (IV) *in/out*. After both experts had finished the labeling, the labels were checked again by expert 1 using visual inspection, see Figure 4, and corrected if necessary. Using this labeling methodology, we aimed to obtain as precise as possible annotations with human annotators, where some degree of human error and mislabeling cannot be completely ruled out. Especially in the game phase, activities are often performed both quickly and briefly, which can lead to minor deviations in labels between manual annotations.

Specifically, (I) coarse separates the samples into different sessions, including (1) *warmup*; (2) drills: (a) *sitting*, (b) *standing*, (c) *walking*, (d) *running*, (e) *dribbling*, (f) *penalty_shots*, (g) *two_point_shots*, and (h) *three_point_shots*; (3) *game*; and (4) *in/out*. By keeping the information whether a shot is either a (f) *penalty_shots*, (g) *two_point_shots*, or (h) *three_point_shots* later studies can use these labels to distinguish between different shot distances. The label (3) *game* indicates when a game was played. The study from Germany contains 2 game sessions with ~10 min each and the study conducted in the USA contains one session of ~22 min. The two layers (II) *basketball* and (III) *locomotion* contain the labels that correspond to one of the classes shown in Table 7, as well as the label *not_labeled*, which is used whenever the information of what exactly a player is doing at a specific moment, could not be seen in the ground truth video or between sessions. The fourth layer in/out is only relevant during the game session since this layer indicates whether a player is on the court or not. However, this layer can be seen as additional meta-information, which can be relevant for future researchers. It has not been used for deep learning validation, since the challenge of classifying if someone is active or non-active seems to be trivial in this scenario.

**Class Definitions:** The following Table 7 contains the class descriptions and Figure 5 visualizes one example for each class.

Sitting and standing mostly show sedentary acceleration patterns with sporadic movements of people moving their wrists, while walking and running show the commonly known oscillating patterns. Dribbling can vary depending on how a player is dribbling. For example, a player can dribble with their dominant or non-dominant hand, dribble the ball by switching hands, or do even fakes and tricks. These styles have slightly different characteristics and can be distinguished, see [67]. However, we decided to summarize these differences in one class.

Even when the ball is dribbled with the non-dominant hand, the data from the dominant hand shows the oscillating characteristics of the dribbling movement. Jumping is an assembled class that also includes jumps belonging to either a shot, rebound, or layup activities. These classes share the trait that the jump—a peak on the coronal plane—is clearly visible. However, the classes differ mainly in the sensor data prior to the peak. The shot contains the player grabbing and lifting the ball before jumping mostly straight up to shoot or, in the case of a penalty shot, performed in a standing position. A rebound is mainly a clear jump upwards or in the forward direction and a layup contains the combination of running 2 or 3 steps (depending on FIBA or NBA rules), a jump, and throwing the ball in the basket while jumping forward. The pass is a very short activity characterized by a forward acceleration on the sagittal plane. Figure 6 shows how the classes are distributed over the sessions.

As one can see, locomotion activities such as *walking* and *running* are distributed almost equally over the sessions. *Sitting* was not performed during the warm-up session and *layup* is almost exclusively performed during warm-up and the game. Samples labeled as *dribbling* and *shot* were mostly recorded during the drill sessions and, similar to layups, performed way less in the game. *Rebound* is the least recorded activity. The imbalance is caused by the realistic recording setup of the dataset and reflects the reality of a training session including training games in basketball. The imbalance should be considered a challenge rather than an obstacle since all data recorded in real environments show such characteristics. Most of the datasets mentioned in Table 1 share an imbalance either with regards to the class distribution or study participant homogeneity. Further, we believe that future studies would benefit from (rather than negatively impacted by) class imbalance and intersubject variability in the dataset. Even though evaluation metrics may not reach their maximum easily, we argue that this setup is more realistic and more representative of a recreational sport itself and will help researchers to understand open research questions better than a fully balanced dataset.

**Combining Classes:** The layers provided in our dataset make it possible to extend it with additional and more challenging classes. For example, shots can be distinguished between *penalty_shots*, *two_point_shots*, and *three_point_shots* by taking into account the *coarse* layer. The *locomotion* layer holds the information if the activity *dribbling* was performed while the player was *standing*, *walking*, or *running*. Therefore, the class definitions in Table 7 only contain the basic classes and can be extended individually—depending on the requirements of one’s project.

## 5. Analysis

This section will provide a preliminary inspection of our dataset. The range of methods employed here includes descriptive statistics, baseline statistical analyses, and machine learning performance results. Our feature analysis focuses on experts vs. novices since we believe that this feature is a strong asset of our dataset that needs to be highlighted. Differences in the data with regard to the players’ experience are visible through features and can be used in later research to develop systems that react to these differences, such as supporting and accompanying a player in the further development of his/her playing skills. If researchers would like to use the dataset as a benchmark dataset for deep learning experiences, they can exclude one or the other group. Exemplary for the rest of the dataset, we included six players in our analysis, with three players from each subset, analyses of the remaining subjects are in Appendix A, where these differences are apparent.

### 5.1. Feature Analysis

Our analysis focuses on the representation of intraclass variability and interclass similarity, as well as clarifying the differences between novices and experts. Figure 7 contains the raw data of approximately 7 min of dribbling while the person stands, walks, and runs with different velocities. The locomotion speed increases over time. Through visual inspection, we can already clearly see that the dribbling patterns differ greatly between novices and experts.

The results of the Fast Fourier Transform (FFT), visible in column (3), indicate that expert players dribble the ball with a wider frequency spectrum than novices, caused by variations in the dribbling style (changing hands, dribbling low/high or fast/slow, or doing tricks). Furthermore, the expert players show a higher mean frequency as well as a higher magnitude column (4) than novice players. Exceptions from this only occur in the feature analysis of players *b512_na* and *0846_eu*, see Figure A4 in Appendix A. However, this is explainable since player *b512_na* mostly dribbled the ball at a walking pace (visible in the video footage) and player *0846_eu* has intermediate dribbling skills, even though the overall skill level can be categorized as a novice. Additionally, for the features depicted in Figure 7, we calculated the arithmetic mean of dribbles per second (AM D.) and the Signal-to-Noise Ratio (SNR), as defined in the following equation.
(1)$$SNR_{db}=10\cdot log_{10}\left(\frac{Psignal}{Pnoise}\right)$$

The experts have a higher rate of dribbles per second than the novices, and this shows that experts dribble the ball more comfortably resulting in fewer ball losses and a faster pace, shown in column (4) of Figure 7. More significance is illustrated in the SNR between the two groups. A higher value means that more noise is present in the signal, or the ball is dribbled in a less controlled manner, and this is also visible in the raw data of Figure 7.

The Principle Component Analysis (PCA) [101], shown in Figure 8, calculated for the same participants shows, exemplary on the basis of the classes *shot* and *layup*, the intraclass variability but also the interclass similarity mentioned at the beginning. The first column contains all subjects, the following three columns contain the experts, and the last three columns contain the novices. The PCA shows that novices follow less coherent movement patterns. Experts, however, present more similar patterns, which differ minimally on both component axes of the PCA.

The following Figure 9 and Figure 10 show 10 examples for the same six participants used in the figures before. The shots show participant-independent patterns that include a negative peak on the z-axis (jump) followed by a positive peak on the y- and z-axis (shot). Such a coherent pattern is hardly visible for the *layup* class.

Such perfect examples, as seen in Figure 5 are rare for the *layup*, especially when the player is contested. The intensity, as well as the execution of the activity, differs a lot depending on the situation.

### 5.2. Deep Learning Analysis

We investigated a variety of prediction scenarios to provide a first impression of potential test cases and benchmark scores that can be achieved using the Hang-Time HAR dataset. As architectures for our classifiers, we chose to use both a shallow variant of the DeepConvLSTM network [102] and Attend-and-Discriminate network [103]. Each of the defined training scenarios employs either a Leave-One-Subject-Out (LOSO) cross-validation or a train-test split to evaluate a network’s predictive performance. The former (LOSO) involves each subject becoming the validation set once while all other subjects are used for training the network, while the latter (split), as the name suggests, simply splits the data into two parts—one used solely for training and the other used solely for testing. During all experiments, we employ a similar hyperparameter setup as used in [102]. We further alter the architecture suggested by Abedin et al. [103] to encompass the findings discussed in [102], i.e., employing a one-layered recurrent part and utilizing 1024 hidden recurrent units for both architectures. Lastly, in order to minimize the effect of statistical variance, for each test case we calculate the average predictive performance across three runs, each time employing a different random seed drawn from a predefined set of three random seeds. In order to determine a suitable sliding window size, three different window lengths, i.e., 0.5, 1, and 2 s, with an overlap of 50% were evaluated using a LOSO cross-validation on the complete Hang-Time HAR dataset. Among the tested window lengths, the results showed little to no difference with the standard deviation of the macro F1-score ranging only between 0.4% (shallow DeepConvLSTM [102]) and 1.2% (Attend-and-Discriminate [103]). Nevertheless, we determine a sliding window length of 1 s with an overlap ratio of 50% to be most suited for the Hang-Time HAR dataset as we expect that:A smaller window length would not be able to capture enough data, and thus patterns specific to activities, which could be learned by the network.A larger window length would capture too much data, increasing the risk of patterns specific to short-lasting activities being mixed with patterns of other activities. This would make it less likely that a network learns to attribute only relevant patterns to short-lasting activities.


During our experiments, we are investigating how well our network generalizes in two regards:



*Subject-independent generalization:* As with almost any activity, basketball players tend to have their own specific traits in performing each basketball-related activity. Within these test cases, we investigate how well our network generalizes across subjects by performing a LOSO cross-validation on the drill and warm-up data of all subjects. During each validation step, the activities of a previously unseen subject are predicted, and thus the experiments will determine how well our network generalizes across subjects and whether subject-independent patterns can be learned by our architecture.*Session-independent generalization:* As previously mentioned, data recorded during an actual basketball game can heavily differ from "artificial" data recorded during the drill and warm-up sessions, as subjects did not have to adhere to any (experimental) protocol. Thus, the session-independent test cases investigate how well our network predicts the same activities performed by already-seen subjects during an actual game. Within these experiments, we train our network using data recorded by all subjects during the drill and warm-up sessions and try to predict the game data of said subjects. These type of experiments will give a sense of how well our network is able to generalize specifically to real-world data and simulates the transition from a controlled to an uncontrolled environment. The network learns player-specific patterns from the warm-up and drill sessions and tries to classify the more dynamic game subset.


In the following, the results obtained during the two test case types will be illustrated. All results as well as the raw log files of each test case can be found on the projects’ Neptune.ai page [104]. The used architectures and the code of the performed experiments are published in our GitHub repository [105].

Looking at the results in Figure 11 and Figure 12 one can see that there are major differences regarding how well our network generalized across study sessions (parts) and across subjects. Overall one can see that using the Hang-Time HAR dataset as input both architectures did not generalize well across sessions, i.e., from drills to games. Looking at the subject-independent results one can see that almost all classes tend to transfer well with only layups (<45% macro F1-score), passing (<30% macro F1-score), and rebounds (<10% macro F1-score) as outlying activities, with the average macro F1-score above 50% for both architectures. Contrarily, the session-independent results show a significant decrease in overall predictive performance by around 24% for the shallow DeepConvLSTM and around 19% for the Attend-and-Discriminate architecture. Nevertheless, this trend does not apply to all activities equally, with most locomotion activities (walking, running, and sitting) not as heavily affected (<50% macro F1-score) in prediction performance as the basketball activities (dribbling, shooting, passing, rebound, and layup) whose macro F1-scores do not exceed 20% for both architectures. We accredit this drop in performance to the fact that basketball games by nature have more unforeseen situations to which players need to adjust their movement too. In general, it is rare that players are able to perform, e.g., an uncontested layup (e.g., certain fastbreak situations) resulting in altered feet and arm movement in order to find the necessary space and successfully score. The influence of a game-like situation can particularly be seen in the locomotion activity standing which sees a major decrease when trying to be predicted in-game. Players constantly move to defend an oncoming player of the opposing team, which makes standing in-game very much different from standing during drills, as players are, e.g., going into a defensive position or are keeping in contact with their assigned player on defense.

We identify the challenges for future research and experiments to be two-fold:Results obtained during session-independent experiments show the poor generalization of basketball-related activities from controlled to uncontrolled environments. This further underlines the bias introduced by researchers when relying on data recorded in a controlled environment compared to uncontrolled environments. It should be investigated whether it is possible to increase generalization through means of altering the training process or employing architectures.Employing the definition as defined in [106], Hang-Time HAR offers both complex (shot, layup) and sporadic (rebound, pass) activities. As these activities are not as reliably detected (even in controlled an environment) as other activities, it is to be investigated whether this lies in the nature of the activities, or can be accredited to the employed network architecture reaching its limits.

## 6. Discussion

We present our dataset Hang-Time HAR, an extensive dataset for (Basketball) Activity Recognition. The dataset was recorded in two different sessions and continents (using two different sport-specific rule sets) in a real-world scenario with approximately 2266 min of real basketball training sessions and training games. The dataset we introduce offers a large variety of activities performed by 24 subjects in both (partly) scripted (drill and warm-up) and unscripted (game) recording sessions. Activities range from simple ones, such as a player‘s locomotion, to complex ones, such as layups and shooting which consist of in-activity sequences. Each basketball player was equipped with a single wrist-worn inertial sensing smartwatch, and labeling was performed by annotating video footage of the sessions.

The feature analysis shows that Hang-Time HAR has considerable intraclass variability and interclass similarity as described by Bulling et al. [13]. This effect was strengthened by the recording setup of a semi-controlled environment. From the perspective of deep learning for human activity recognition, the dataset offers a variety of new challenges. As evident in the results of our Deep Learning analysis, during the LOSO cross-validation the architectures we chose reached their limits with respect to the classes *rebound* and *layup* in both session types we evaluated, see Figure 11 and Figure 12. To be able to recognize it in a LOSO cross-validation, where no prior information on the subject is given to the classifier, we need either more samples of that class, e.g., through applying techniques such as data augmentation or a deep learning architecture that is able to handle under-represented classes. Basketball-specific classes were predicted during the game, on average, with a 25% F1-Score. *Passes* and *rebounds* were extremely difficult for the classifier to detect since their execution time is often under 1 s. Furthermore, the most significant part of the activity *rebound* is the jump—which is a sub-activity that is shared with other classes such as *shot* or *layup*. These activities that were predicted poorly when missing subject-specific training data also correspond to the fewest samples in the dataset. Future work could involve testing techniques such as artificially increasing the under-represented classes through data augmentation or a more suitable deep learning architecture for handling class imbalances.

According to Bock et al. [106] we distinguish between sporadic, simple/periodical, transitional, and complex activities. However, datasets shown in Table 1 mostly focus on locomotion activities and activities of daily living. Only a few, such as [26,27,35,51], include sporadic, transition or complex activities, and many datasets that do include sports [16,33] aggregate an entire sport into a single activity. Published sports studies tend to not release their datasets publicly or only upon request—with Trost et al. [17] and Bock et al. [51] as the only exceptions, as shown in Table 2 and Table 3. As a result, sports-specific IMU-based datasets available to the public that reflect the complexity and characteristics of a specific sport are very limited. Due to the nature of the sport of basketball, our dataset contains classes where the characteristics mentioned by Bock et al. apply. *Rebound, pass, and jump* can be considered as sporadic classes. The locomotion classes—*sitting*, *standing*, *walking*, and *running* are periodic classes, *shot* and *layup* contain complex, interrelated activities. During the warm-up and game, all activities were situation-based and therefore the dataset contains natural and fluently performed transitions between classes as well as overlapping activities.

By using the different semantic layers of the dataset—*coarse*, *basketball*, *locomotion*, and *in/out*—researchers are able to focus on different aspects of activity recognition by studying the different semantic levels either in isolation or in combination and incorporate them in their research appropriately. In particular, the combination of various semantic levels offers researchers the possibility to design and develop game analysis algorithms based on IMU signals. Such algorithms could either analyze the players’ performance with or without focusing on specific activities or could analyze the game itself in a holistic approach. Wrist-worn smartwatches—as used in this study—are not allowed to be worn during an official basketball game, since they bear the risk to harm the player or other players on the field. However, we believe that it can be replaced in future studies, e.g., by sweatbands that incorporate IMU sensors. Such a device could come in a similar form as those in [107] or [78].

Apart from providing a benchmark dataset for future machine and deep learning studies, we believe that our dataset has cross-domain application purposes that can help to solve open research questions such as activity recognition of complex classes in real-world scenarios, including the development of preprocessing or postprocessing algorithms for real-world data as well as designing neural network architectures for such scenarios. In particular, the game data introduces a completely new scenario for human activity recognition in which activities overlap each other and are performed with a higher pace and altering patterns, due to the ball possession and the psychological pressure during a game situation. Such semantic learning can become another important sub-field for HAR in the near future, as demonstrated by the recently published architecture SemNet by Venkatachalam et al. [108].

It is known that transfer learning for HAR does not perform equally well as it does for vision data. Pretraining and transferring a neural network do have a positive effect on the classifiers’ capabilities as well as the training time [109]. Our dataset can help explain these phenomena since the locomotion layer is class-wise compatible with many other datasets shown in Table 1 and should be therefore transferable. However, due to the recording environment and activity domain, the classes are expected to differ from similar classes published by datasets shown in this table. Further transfer learning studies that test the effectiveness of pretraining a neural network model with regard to several domain-specific datasets can be of interest to the activity recognition community. We think that pretraining a neural network on a sports dataset and transferring the model to another sport can have a higher positive impact on the classifier than pretraining it on a non-domain dataset. However, this is speculative at this point and needs to be investigated by future studies. The different skill levels of our participants shall not be seen as a disadvantage, but rather as a unique feature that opens up challenges and opportunities for not yet addressed research questions. The distinction between the two levels of skills can help understand the real effect of noisy real-world performed instances of activities on a trained classifier. As Section 5.1 describes, we were able to identify differences in the patterns between *novices* and *experts* due to unclean performed activities, such as shooting the ball with both hands or dribbling the ball with less control than experienced players. Including or excluding one or the other will have an effect on the classifier. We think that these effects are valuable and should be investigated as part of a larger and more complex study in the context of classifier poisoning, transfer learning, or research problems with regard to data labeling.

An IMU-based approach has the advantage over vision-based approaches since wearables (e.g., smart watches) are low-cost, widely available, and quickly deployable to players on every court (indoor and outdoor). Vision-based approaches require a more complex tech build, which is cost- and labor-intensive to set up and configure. Furthermore, in future works, a simple model can be trained and deployed on a wearable device in order to classify motions through IMU data in real-time and on-device. Recent advances, such as the TinyHAR [110], are capable of detecting human activities with fewer trainable parameters and are therefore power-efficient enough to be deployable on wearable devices, such as the Bangle.js smartwatch (the Bangle.js comes with TensorFlow Lite preinstalled on the microcontroller). We, therefore, expect that our benchmark dataset will have a significant impact on activity recognition research in itself, but also encourage more follow-up work in the methodologies for designing, recording, and annotating such datasets. We argue that the sports domain, in general, offers researchers a recording environment that can range from a controlled to an uncontrolled setting and with the advantage that data can be labeled retrospectively using video footage.

The players’ meta information can be used to gain deeper insights into how a person’s build and sports experience affects the execution style of an activity. Even though the metadata contains basic information about the players’ prior basketball experience, it does not claim to evaluate the playing skills of individual players. The video footage might be used to provide such annotations to be added as extra annotation layers. In future work, we would like to add another layer to the game sessions called def/off, which indicates whether a player is currently playing defense or offense, respectively. This information is useful with regard to a player’s locomotion since the defense position is usually played in an upright position with hands raised and knees slightly bent, see Figure 1. Furthermore, since this paper focuses on feature analysis and deep learning driven classification methods, medical statistics such as a Bland–Altman analysis [111] and biomedical derived studies [65,112,113], fall out of our scope of study but could supplement this paper in future works.

Besides the use case of basketball activity recognition, we expect that certain activities are generalizable across different sports. Periods in which a player did run without dribbling (the periods can be filtered by taking into account the different semantic levels of annotations) can be transferred to sports such as handball, indoor soccer, futsal, or in general indoor sports that share similar field size and have periods of players running without a ball. The dribbling movement seems to be transferable between basketball and handball. However, we expect that the transferability will have its limitations. For example, the class jumping will have different characteristics in volleyball compared with a jump in basketball, since the game volleyball itself is more focused on the vertical space and has different patterns depending on what action the players perform. Volleyball has basically six different skills that players perform during a game, which are: serve, pass, set, attack, block, and dig. All of them, except two special variants of serve and pass (float serve and forearm pass) involve jumping. Hypothetically, if a wrist-worn sensor-based dataset with volleyball activities would be published, it would be interesting to explore whether the class jumping would be transferable.

## 7. Conclusions

During this study, we have developed a basketball activity dataset that brings a variety of unique features with it and is, to the best of our knowledge, the only sensor-based and publicly-available activity recognition dataset that focuses on fine-grained team sports activities. The dataset introduces data from wrist-worn inertial sensors of 24 players from two teams and recorded in two different continents where slightly different rule sets are applied. The participants perform ten different basketball activities that are grouped into four different semantic levels. The dataset contains warm-up, drills, and game phases. Typical routines were followed during the drills but not during the warm-up and game, where players were allowed to play as they preferred. Therefore, this dataset contains data from controlled as well as uncontrolled environments which can be filtered as needed by researchers. The different semantic levels of the annotations make it not only possible to focus on general locomotion or specific basketball activities, but also to create more complex classes as mentioned before. The two levels of skills, novice, and expert, inherit a strong intraclass and intersubject signal-variability which has already been mentioned by Bulling et al. [13] in 2014 and is still an ongoing research challenge in real-world scenarios. Therefore, we argue that this feature is directly relevant to real-world activities of any domain and can be used to investigate these problems further. As aforementioned, the class *not_labeled* contains data that corresponds to NULL activities as well as activities that are not part of the dataset. As such, this class represents a very realistic and naturally-designed *void* class which can be of interest to studies that focus on investigating the NULL-class problem. The results of our deep learning analysis show that current architectures are not capable of detecting complex classes. In order to overcome this obstacle, further research on data preprocessing and architectural neural network design is needed. This problem becomes more challenging if the data is recorded in a real-world and uncontrolled environment.

Given the uniqueness as a fine-grained sports dataset, the class variability, the high number of study participants and therefore resulting size of the dataset, and the comprehensive coverage of rule-set-varying characteristics of the sport of basketball, we firmly believe that this dataset will be suitable for the evaluation of machine learning and deep learning algorithms, network architectures, and previously mentioned problems and will establish itself as a benchmark dataset for the human activity recognition community across application domains.

## Figures and Tables

**Figure 1 sensors-23-05879-f001:**
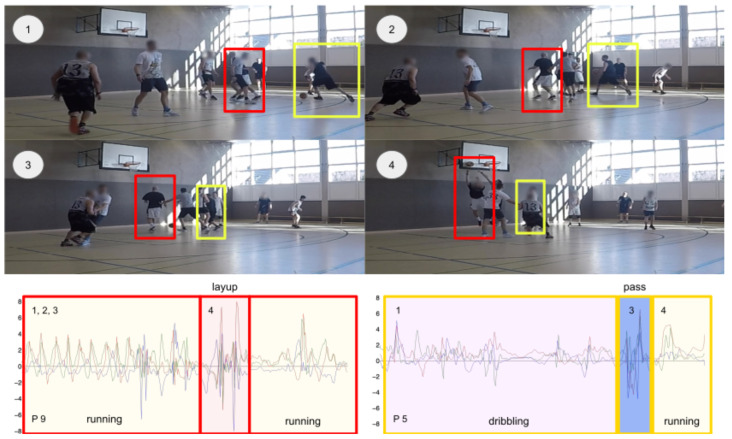
A scene and activities from the dataset: Offensive play of player 12 (yellow) and player 6 (red), see Table 6, with player 12 dribbling the ball (1), (2) and then passing (3) it to player 6. Player 6 then performs a layup (4). Video frames 1–4 and the performed activities are highlighted in the time-series below. The activity *running* is marked as yellow, *layup* as red, *dribbling* as mauve and *pass* is colored in blue.

**Figure 2 sensors-23-05879-f002:**
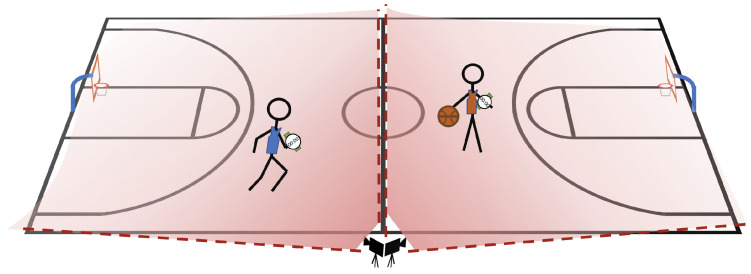
Our study design used 24 subjects with 13 subjects living in Germany and 11 subjects living in the United States of America. In each study, the players simultaneously performed the drills and game while the entire basketball court was monitored using two wide-angle cameras. After the study, the camera footage was used for detailed annotation of all activity-relevant data.

**Figure 3 sensors-23-05879-f003:**
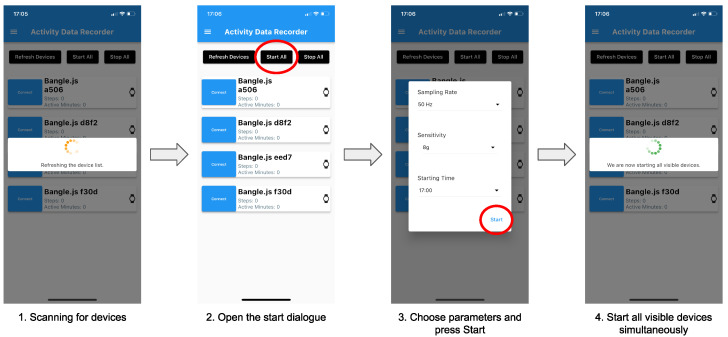
Our custom smartphone app was used to synchronize all smartwatches’ real-time clocks at the beginning of each recording through Bluetooth Low Energy (BLE) serial commands and start recording simultaneously. After the app is started, it first scans for all available Bangle.js smartwatches. After that, the user has the option of either starting all devices simultaneously or individually. Before the smartwatches are started, the user is asked to enter the desired parameters (sampling rate, sensitivity, and start time). After pressing the start button, all smartwatches are started with the desired parameters.

**Figure 4 sensors-23-05879-f004:**
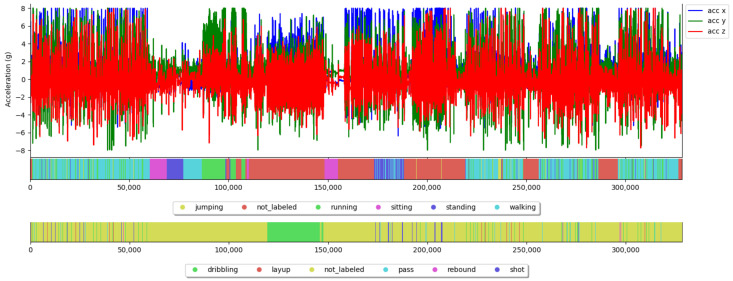
Illustration of the multi-tier labeling approach, depicting the inertial data of subject *05d8_eu* (top), the ground truth locomotion Layer (middle), and the ground truth basketball layer (bottom).

**Figure 5 sensors-23-05879-f005:**
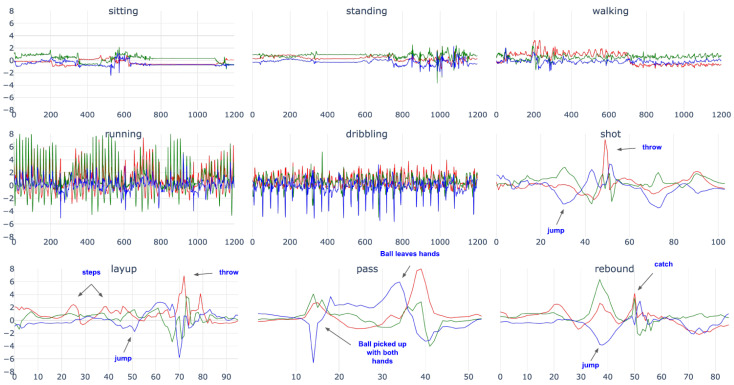
Exemplar time-series data for the included activities. The examples shown for the periodic activities *sitting*, *standing*, *walking*, *running*, and *dribbling* contain 1200 samples (approx. 24 s). In order to better represent the complex activities *shot* and *layup* as well as the micro-activities *pass* and *rebound*. Jumps are marked in classes where the activity occurs. Such short periods were summarized in the activity *jumping*.

**Figure 6 sensors-23-05879-f006:**
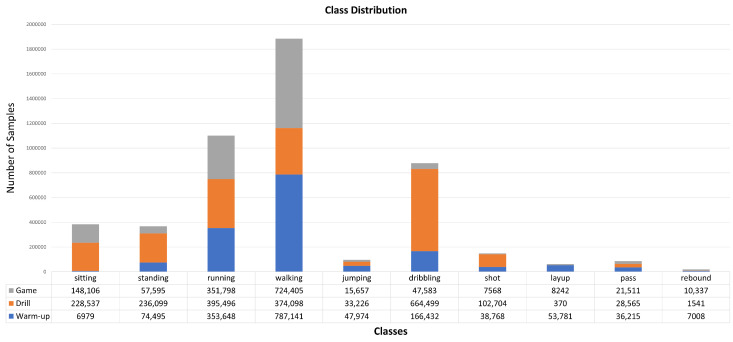
Class distribution of the Hang-Time HAR dataset. Total number of samples per class are: *sitting*: 383,622 (~2.1 h), *standing*: 368,189 (~2.0 h), *walking*: 1,885,644 (~10.5 h), *running*: 1,100,942 (~6.1 h), *jumping*: 96,857 (~0.53 h), *dribbling*: 878,514 (~4,8 h), *shot*: 149,040 (~0.82 h), *layup*: 62,393 (~0.34 h), *pass*: 86,291 (~0.47 h), and *rebound*: 18,886 (~0.10 h). In total: 5,030,378 labeled samples or ~27.7 h of data.

**Figure 7 sensors-23-05879-f007:**
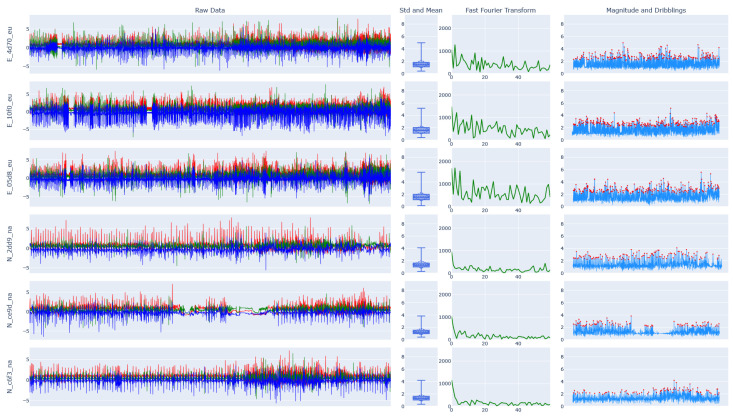
Feature analysis of the class dribbling for players 4d70_eu, 10f0_eu, and 05d8_eu (experts) and 2dd9_na, ce9d_na, and c6f3_na (novices). The plot consists of 4 columns. (1) Raw data as recorded during the dedicated dribbling drill (approx. 7 min of data (Germany) and 5 min of data (USA). The X-axis is represented in red, the Y-axis in green, and the Z-axis in blue color. (2) Standard deviation (diamond shape), median, interquartile q1 and q3 (rectangle shape) as well as upper and lower fences. (3) Fast Four Transformation [99]. (4) Local maxima [100] (*prominence = 1.4*) calculated using the magnitude of the input signal (1), every red dot indicates a peak that is interpreted as one dribbling.

**Figure 8 sensors-23-05879-f008:**
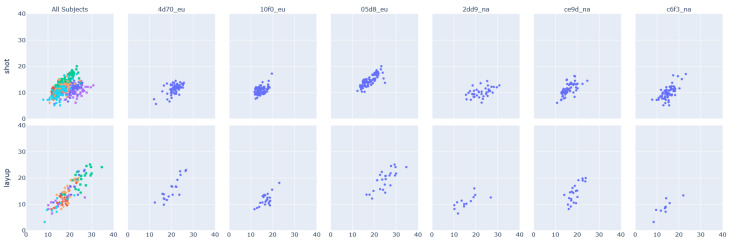
Principle Component Analysis of the classes (1) shot and (2) layup. For the same subjects mentioned in Figure 7 and Table 8. The colors represent the 6 different participants included in this figure. *4d70_eu* is represented in blue, *10f0_eu* in red, *05d8_eu* in green, *2dd9_na* in purple, *ce9d_na* in orange, and *c6f3_na* in turquoise.

**Figure 9 sensors-23-05879-f009:**
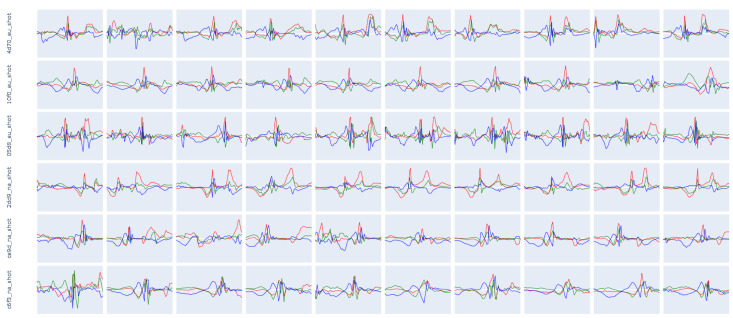
Ten instances of the class *shot* for the same subjects as mentioned in Figure 7 and Figure 8. A clearly visible pattern can be seen in all examples. The length of the activity typically varies between 1000 and 3000 ms with an average duration of approx. 1700 ms, depending on the subject and the execution style. The X-axis is represented in red, the Y-axis in green, and the Z-axis in blue color.

**Figure 10 sensors-23-05879-f010:**
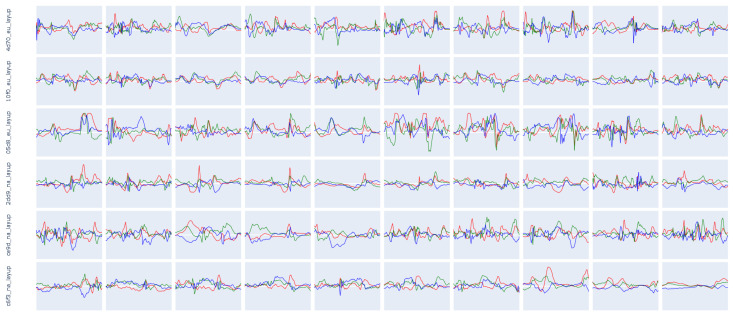
Ten instances of the class *layup* for the same subjects as mentioned in Figure 7 and Figure 8. The patterns vary by subject and sometimes even differ between instances of the same subject. This class inherits a strong intraclass and intersubject variability. The length of the activity typically varies between 1000 and 3000 ms, with an average duration of approx. 2000 ms, depending on the subject and the execution style. The X-axis is represented in red, the Y-axis in green, and the Z-axis in blue color.

**Figure 11 sensors-23-05879-f011:**
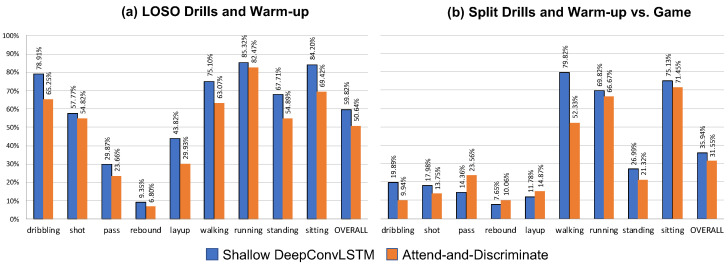
Overall results of the deep learning experiments using a shallow DeepConvLSTM (blue) and Attend-and-Discriminate architecture (orange). Both models were trained with a 1-layered recurrent part with 1024 hidden units and a sliding window of 1 s with 50% overlap. The left plot (**a**) shows the per-class LOSO results obtained from training on the drill and warm-up data. The right plot (**b**) shows the per-class results predicting the game data when trained on the drill and warm-up data. All results are averages across 3 runs using a set of 3 random seeds. Both architectures suffer a significant loss in predictive performance when being applied to in-game data, i.e., data recorded in an uncontrolled environment.

**Figure 12 sensors-23-05879-f012:**
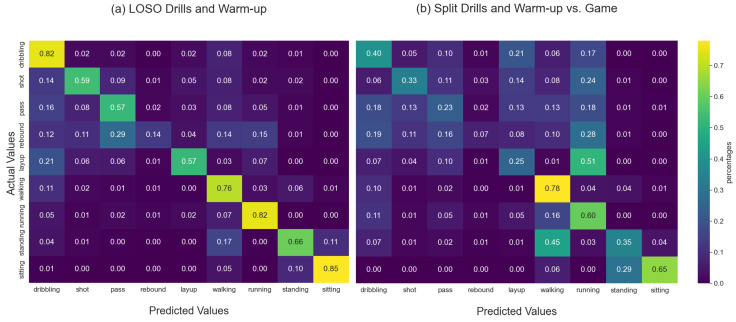
Confusion matrices of a shallow DeepConvLSTM applied to the Hang-Time dataset. The model was trained with a 1-layered recurrent part with 1024 hidden units, a sliding window of 1 s with 50% overlap, and a fixed random seed. The left confusion matrix (**a**) is obtained from averaging the per-subject LOSO results using the drill and warm-up data as input data. The right confusion matrix (**b**) is obtained from training on the drill and warm-up data and validating on the game data. One can see an increase in overall confusion when applying the architecture to in-game data, i.e., data recorded in an uncontrolled environment.

**Table 1 sensors-23-05879-t001:** The most relevant datasets used by the HAR community, as well as examples for datasets from uncontrolled or semi-controlled environments (with challenges based on Table 1 from [6]). Our presented dataset Hang-Time HAR is highlighted.

Dataset	Device	# Subjects	# Classes	Domain	Environment	Challenges	Published
HHAR [24]	Smartphone	9	6	Locomotion	Controlled (Lab)	Multimodal, Distribution Discrepancy	2015
RWHAR [25]	Smartphone, Wearable IMUs	15	8	Locomotion	Controlled (Outside)	Multimodal	2016
Opportunity [26]	Wearable IMUs, Object-Attached Sensors, Ambient Sensors	4	9	ADL, Kitchen Activities	Controlled (Lab)	Multimodal Composite Activity	2010
Opportunity++ [27]	Wearable IMUs, Object Attached Sensors, Ambient Sensors	4	18	ADL, Kitchen Activities, Video, OpenPose tracks	Controlled (Lab)	Multimodal Composite Activity	2021
PAMAP2 [28]	Wearable IMUs	9	18	Locomotion, ADL	Controlled (Lab, Household)	Multimodal	2012
Skoda [29]	Wearable IMUs	1	12	Industrial Manufacturing	Controlled (Industrial Manufacturing)	Multimodal	2008
UCI-HAR [30]	Smartphone	30	6	Locomotion	Controlled (Lab)	Multimodal	2013
WISDM [31]	Wearable IMUs	29	6	Locomotion	Controlled (Lab)	Class Imbalance	2011
UTD-MHAD [18]	Wearable IMUs, Video	8	27	Gestures, Sports	Controlled (Lab)	Multimodal	2015
Daphnet [32]	Accelerometer	10	3	ADL, Locomotion	Controlled (Lab)	Simple	2009
DSADS [16]	Wearable IMUs	8	19	Sports, ADL	Controlled (Lab & Gym)	Multimodal	2010
ActiveMiles [23]	Smartphone	10	7	Locomotion	Uncontrolled (In-The-Wild)	Real-World	2016
Baños et al. [33]	Wearable IMUs	17	33	Sports (Gym)	Controlled (Gym)	Multimodal	2012
Leisure Activities [15]	Wearable IMU	6	6	ADL	Uncontrolled (In-The-Wild)	1 activity per subject	2012
WetLab [34]	Wearable IMU, Egocentric Video	22	9	Experiments (Wetlab)	Semi-Controlled (Wetlab)	Multimodal	2015
TNDA-HAR [19]	Wearable IMUs	23	8	Locomotion	Controlled (Lab)	Multimodal	2021
CSL-SHARE [35]	Wearable IMUs, EMG, Electrogoniometer, Microphone	20	22	Locomotion, Sports	Controlled (Lab)	Multimodal	2021
**Hang-Time HAR**	**Wrist-worn** **accelerometer**	**24**	**15**	**Sports (Basketball)**	**Controlled and** **uncontrolled (Gym)**	**Different recording** **environments,** **Class Imbalance**	2023

**Table 2 sensors-23-05879-t002:** IMU-based studies have been performed throughout many different sports in the past years, yet few are publicly available for usage by other researchers (table partially based on [58], Table 2).

Sports Studies with Wearables					
**Study**	**Sport &** **(#) Activities Performed**	**Sensors/Systems Used**	**# Subjects**	**Dataset Published**	**Analysis Method**
Bastiaansen et al. [37]	(1) Sprinting	Five IMUs and sensor fusion algorithms	5	No	Statistical Analysis
Borja Muniz-Pardos et al. [38]	(1) Running	Foot worn inertial sensors	8	No	Statistical Analysis
Brouwer et al. [59]	(5) Swing motions from different sports: golf swi- ngs, 1-handed ball throws, tennis serve, baseball swings. and a variety of trunk motions.	Two IMUs and a MoCap system	10	No	Statistical Analysis
Brunner et al. [60]	(5) Swimming	Wrist-worn full IMU, barometer	40	No	Deep Learning (CNN)
Carey et al. [42]	(1) Physical impacts while playing rugby	head-worn accelerometer and gyroscope (x-patch™)	8	No	Statistical Analysis
Lee et al. [45]	(2) Skiing turns	17 IMUs and pressure sensors	7	No	3D Kinematic Model Evaluation
Teufl et al. [49]	(3) Bilateral squats, single leg squats, and counter- movement jumps	Seven IMUs and a MoCap system	28	No	Rigid Marker Cluster, Statistical Analysis
Wang et al. [61]	(3) Racket Sports	Wrist-Worn IMU	12	No	Machine Learning, (SVM, Naive Bayes)
Whiteside et al. [62]	(9) Tennis strokes	Wrist-Worn IMU	19	No	Statistical Analysis
Ghasemzadeh and Jafari [41]	(1) Baseball swing	3 IMUs (Wrist, Shoulder, Hip)	3	No	Semi Supervised Clustering
MacDonald et al. [43]	(15) Volleyball	6D IMU (Acc. & Gyr.)	13	No	Statistical Analysis
Borges et al. [44]	(6) Volleyball	Waist worn full IMU	112	No	Statistical Analysis
Dahl et al. [50]	(5) Cutting, running, jumping, single leg squats and cross-over twist	8 full IMUs, 17 MoCap Cameras	49	No	Statistical Analysis
Pajak et al. [48]	(4) Fitness exercises: dips, pullups, squats, void	3 full IMUs, Pressure Sensor, Radio Signal	-	No	Deep Learning (CNN)
Yu et al. [40]	(1) Soccer kick	6D IMU (Acc. & Gyr.)	-	Yes, upon request	Attitude Estimation with Quaternions, Gravity Compensation
Stoeve et al. [36]	(3) Soccer kick, pass, void	Shoe-worn IMU	836	No	Machine and Deep Learning (SVM, CNN, DeepConv-LSTM)
Bock et al. [51]	(19) Fitness activities	4 Accelerometer sensors, egocentric video footage	18	Yes	Deep Learning (DeepConv-LSTM, Attend-and-Discrim- inate, ActionFormer)
Brognara et al. [52]	(-) CrossFit^®^	Full IMU at the lower back	42	Yes, upon request	Statistical Analysis
Perri et al. [63]	(8) Tennis strokes	1 Full IMU at the scapulae	8	Yes, upon request	Statistical Analysis
Azadi et al. [46]	(1) Alpine skiing	2 smartphones with IMUs placed at the pelvis	11	No	Unsupervised Machine Learning (Gaussian Mix- ture Models, Kmeans)
Jean et al. [53]	(-) Running	foot-worn 6-axis IMU	41	No	Statistical Analysis
Yang et al. [56]	(-) Contact and flight-time (Running)	2 ankle-worn 6-axis IMUs	36	Yes, upon request	Statistical and Feature Analysis
Léger et al. [64]	(3) Ice Hockey	1 glove-worn IMU	10	Yes, upon request	Machine Learning (kNN)
Hamidi et al. [54]	(-) Swimming perfor- mance	1 sacrum-worn IMU	15	Yes, upon request	Statistical Analysis, Self-Assessment
Müller et al. [55]	(-) Beach Handball performance	1 full IMU placed at the upper thoracic spine	69	Yes	Statistical Analysis
Patoz et al. [57]	(-) Contact and flight-time (Running)	1 sacral-mounted IMU	100	Yes, upon request	Statistical Analysis
Lee et al. [65]	(4) stride, step, and stance duration of running gait	Sacrum worn 3D Accelerometer, 6 infrared cameras	10	No	Statistical Analysis
Harding et al. [66]	(-) Airtime analysis of snowboarders	One 3D gyroscope	10	No	Statistical Analysis

**Table 4 sensors-23-05879-t004:** Vision-based basketball activity recognition studies that published their dataset for download. However, Maksai et al. and Ramanthan et al. are currently not available for download. (* Not accessible from the source given by the manuscript at the time of writing).

Vision-Based Basketball Studies			
**Study**	**Action Recognized**	**Sensors/Systems Used**	**Dataset Published**
Hauri et al. [79]	Group activities: pick and roll, handoff	Videos and 1D-Accelerometer (only shots, extrapolated from videos)	Yes
Ma et al. [84]	12 atomic basketball actions	RGB-D Video Data	Yes
Shakya et al. [85]	two point, three point, mid range shots (success and fail- ures separately classified)	RGB Video and optical flow data	Yes
Gu et al. [80]	3 broad categories: dribbling, passing, shooting; 26 fine-grained actions	RBG Video Data	Yes
Francia [86]	walk, no action, run, defense, dribble, ball in hand, pass, block, pick, shot	RGB Video Data	Yes
Parisot et al. [87]	player detection	RGB Video Data	Yes
De Vleeschouwer et al. [81]	Throw, Violation, Foul Player Exchange, Pass Rebound, Movement	7 cameras, RGB Video Data	Yes, upon request
Maksai et al. [82]	Trajectory estimation	RGB Data of various ball sports (basketball among others)	Yes *
Ramanthan et al. [83]	layups, free throw, 3 point, 2 point shots, slamdunk (success and failures separately classified)	RGB Video Data	Yes *
Tian et al. [88]	basketball tactics detection	RGB Video Data published by [89]	Yes

**Table 5 sensors-23-05879-t005:** Differences between the two studies and a description of the camera recording settings and file sizes for each study and camera employed.

	Ball Regulation	Number of Participants	Study Duration	Video Camera	Duration (Minutes)	Resolution (Pixels)	File Size	FPS	SD Card Capacity
**Germany**	FIBA	13	110	GoPro Hero 4 GoPro Hero 8	110 110	1920 × 1080 1920 × 1080	20 GB 20 GB	60 60	64 GB 64 GB
**USA**	NBA	11	76	GoPro Hero 8 Sony NEX6	76 40	2704 × 1520 1920 × 1080	26 GB 5 GB	60 60	125 GB 32 GB

**Table 6 sensors-23-05879-t006:** Meta information as given through the study questionnaire by all participants, 13 from Germany, Europe (eu) and 11 from USA, North America (na). A total of 3 participants were female and 21 were male. The players were between 18 and 39 years old. Through self-assessment, in which participants were asked to evaluate their experience in basketball, 8 players responded with novice and 16 with expert. Two people were left-handed. Additional about the anthropomorphy of our participants are excluded due to restrictions given by the Ethical Council of our university. **Note:** Subject 2dd9_na wore the smartwatch on the left wrist even though the right hand is dominant.

**Germany**													
**#**	**1.**	**2.**	**3.**	**4.**	**5.**	**6.**	**7.**	**8.**	**9.**	**10.**	**11.**	**12.**	**13.**
**ID**	**e90f_eu**	**b512_eu**	**f2ad_eu**	**4991_eu**	**9bd4_eu**	**2dd9_eu**	**ac59_eu**	**05d8_eu**	**a0da_eu**	**10f0_eu**	**0846_eu**	**4d70_eu**	**ce9d_eu**
**Age**	25	39	20	28	19	34	29	19	20	35	18	36	25
**Dom. Hand**	right	right	left	right	left	right	right	right	right	right	right	right	right
**Height (cm)**	191	167	178	188	190	196	190	178	193	172	171	188	175
**Weight (kg)**	85	85	67	100	80	83	83	77	87	73	60	74	73
**Gender**	male	male	male	male	male	male	male	male	male	male	male	male	male
**Experience**	expert	expert	expert	expert	expert	expert	expert	expert	expert	expert	novice	expert	expert
**USA**													
**#**	**14.**	**15.**	**16.**	**17.**	**18.**	**19.**	**20.**	**21.**	**22.**	**23.**	**24**		
**ID**	**b512_na**	**9bd4_na**	**2dd9_na**	**4d70_na**	**c6f3_na**	**f2ad_na**	**a0da_na**	**ac59_na**	**10f0_na**	**0846_na**	**ce9d_na**		
**Age**	27	26	24	26	24	25	28	28	27	30	24		
**Dom. Hand**	right	right	right	right	right	right	right	right	right	right	right		
**Height (cm)**	165	178	175	183	180	170	170	173	154	165	188		
**Weight (kg)**	68	65	84	68	83	69	73	65	49	65	73		
**Gender**	male	male	female	male	male	male	male	male	female	female	male		
**Experience**	expert	novice	novice	expert	novice	expert	novice	expert	novice	novice	novice		

**Table 7 sensors-23-05879-t007:** Detailed class description for every class included in the dataset. The dataset is multi-tier labeled with 4 different layers (I) Coarse, (II) Locomotion, (III) Basketball, and (IV) In/Out. The coarse layer is not listed, since it is meant to indicate to which session an activity belongs. Relevant classes are classes 2–13. However, the classes *in* and *out* were not used in our validation.

All Layers			
**1. not_labeled**	All samples in between sessions, or if it was not possible to recognize the activity in the video (e.g., due to occlusions).		
**In/Out**			
**2. In**	Indicates that the subject is currently actively participating in the game.	**3. Out**	Indicates that the subject is currently not actively partici- pating in the game. This class mostly included sitting or walking.
**Locomotion**		**Basketball**	
**4. sitting**	Sitting on the floor or the reserve bench.	**9. dribbling**	Dribbling while performing one of the following locomotion activities: (3) standing, (4) walking, (5) running.
**5. standing**	Standing still.	**10. shot**	A basketball shot with and without a jump. Included are penalty shots, 2-point and 3-point shots.
**6. walking**	Walking at the average walking speed of a human (4–5 km/h).	**11. layup**	A layup is a complex class that contains: grabbing the ball, making 2 steps, jumping, and throwing the ball in the basket.
**7. running**	Running is a metaclass for all velocities of running. Therefore, it contains jogging (5–6 km/h), fast running (6 km/h < 10 km/h) and sprinting (>10 km/h).	**12. pass**	Passing the ball. Included are chest passes, bounce passes, overhead passes, one-handed push passes and so-called baseball passes.
**8. jumping**	A jump typically is part of a more complex activity, such as (10), (11), or (13).	**13. rebound**	The player jumps and catches the ball mid-air with one or two hands.

**Table 8 sensors-23-05879-t008:** Arithmetic Mean of dribbles/second (AM D.) and Signal-to-Noise Ratio (SNR) are listed per subject and separated between *experts* and *novices*.

	Experts			Novices		
**ID**	**10f0_eu**	**05d8_eu**	**4d70_eu**	**2dd_na**	**c6f3_na**	**ce9d_na**
**AM D.**	1.10	1.05	1.04	1.01	1.02	1.01
**SNR**	3.40	2.97	3.47	5.93	8.43	7.17

## Data Availability

The full dataset itself and accompanying visualization and experiment tools (written in Python) are available for download at: https://doi.org/10.5281/zenodo.7920485, last accessed on 18 June 2023, and https://github.com/ahoelzemann/hangtime_har, last accessed on 18 June 2023. for replication purposes. The logged experiments can be found at https://app.neptune.ai/o/wasedo/org/hangtime, last accessed on 18 June 2023. The video data was recorded for labeling purposes only and is not part of the published dataset. The IRB waiver/approval of the University of Siegen and the University of Colorado Boulder as well as the informed consent of our participants does not include publishing the video data.

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
