# Peer review of "Hang-Time HAR: A Benchmark Dataset for Basketball Activity Recognition Using Wrist-Worn Inertial Sensors"

_sensors, 2023, doi:10.3390/s23135879_

Round 1
Reviewer 1 Report
This manuscript has presented a benchmark dataset for evaluating physical human activity recognition methods from wrist-worn sensors, for the specific setting of basketball training, drills, and game. The topic of the manuscript is really interesting and timed. The paper is written well, there are research voices in these sections. However, the paper required the following comments and concerns to improve the quality of the presentation.
1-The proposed work should be compared with the relevant studies.
2-What is the main difference between the proposal and others?
3-How this solution outperforms others.
4-The proposal solution should be explained by adding more figures/steps
Done
Author Response
Dear reviewer,
We first of all would like to thank you for taking the time to read our submission and provide us with your valuable expert opinions. In the past week, we have been working hard to address each of your reviews and specific feedback items. In the sections below, we answer how we have updated the submission according to your comments and questions, which accompany our resubmission where changes have been marked as well.
- The proposed work should be compared with the relevant studies.
Thanks for the suggestion. Additional information has been added in Section 3 Related Work on Sports Studies, where we address specifically other basketball studies that are relevant for our presented work and put them in context to our dataset.
- What is the main difference between the proposal and others?
Additional information has been added in Section 3 Related Work on Sports Studies, where we highlight that our dataset contains both complex and periodic classes and consists of controlled (drills and warm-up) and uncontrolled (game) sessions. This is unique for a publicly available dataset.
- How this solution outperforms others.
We have now provided more depth by explaining how our proposed dataset is unique in its (1) scope, (2) quality, (3) variability and (4) complexity. This makes it challenging to estimate to which degree the delivered baseline classifier results might or might not outperform others overall. Important evaluation results that are stated in the paper are that complex and micro activities are difficult to detect when data is recorded under a real setting, such as a sports game, and data from uncontrolled environments can only be transferred up to a certain grade.
The provided baseline results can be used to design and evaluate future classifiers and algorithms that focus on the aforementioned unique characteristics and might outperform our classifiers.
- The proposal solution should be explained by adding more figures/steps
Thanks for the comment. We have reorganized the analysis section, to describe in detail all important aspects of the dataset. Furthermore, we deliver two deep learning classifiers results that other researchers might use as a baseline for future studies.
Reviewer 2 Report
The use of Hang-Time in the title is not supported in the text, as a vernacular term is suggests analysis of air time which is not present in the analysis, nor explained in the text. Suggest removing it, it seems a marketing term only or supporting it further in the text . An example of hang time analysis is in the following
Harding, J. W., Mackintosh, C. G., Hahn, A. G., & James, D. A. (2008). Classification of aerial acrobatics in elite half-pipe snowboarding using body mounted inertial sensors. The Engineering of Sport, 7(2), 447-456.
In many of the figures there is no X axis scale present and often in the same figure the time scale changes, which is left for the reader to infer . Figure 5 shows a figure with multiple scales used, other figures show mixed mode data eg time series and frequency domin without sufficient (or legible labelling)
The review section is commendable however in the Table summaries a further column should be added that describes the methods used by the authors of such papers to improve its utility to the reader
The table with the participate information would benefit through the inclusion of anthropomorphy as this is valuable to many researchers for analysis of data Examples here would be length length and arm length
Section 5 describes various types of analysis however these tend to be solely black box numerical approaches, the paper would benefit by the inclusion of biomechanical derived methods. Where this is not possible stating reasons for exclusion and introducing using papers from the review section or something like
Lee, J. B., Mellifont, R. B., & Burkett, B. J. (2010). The use of a single inertial sensor to identify stride, step, and stance durations of running gait. Journal of Science and Medicine in Sport, 13(2), 270-273.
Ethical clearance is required for the reporting of all data on human subjects. usually the approving institution and a clearance number is reported to ensure data is traceable to its origin and collection methods. Where historical data is used, even if ethics approval was not required by a commercial organisation, my understanding is that ethical clearance to access historical data is still required if it is to be reported in the literature.
Author Response
Dear reviewer,
We first of all would like to thank you for taking the time to read our submission and provide us with your valuable expert opinions. In the past week, we have been working hard to address each of your reviews and specific feedback items. In the sections below, we answer how we have updated the submission according to your comments and questions, which accompany our resubmission where changes have been marked as well.
- The use of Hang-Time in the title is not supported in the text, as a vernacular term is suggests analysis of air time which is not present in the analysis, nor explained in the text. Suggest removing it, it seems a marketing term only or supporting it further in the text.
We decided to keep the name, as it was chosen to reflect the dataset's direct relationship between basketball, time-series data classification and therefore human activity recognition. An explicit explanation on how the name was chosen was also added in our resubmission in Section 2.
- In many of the figures there is no X axis scale present and often in the same figure the time scale changes, which is left for the reader to infer . Figure 5 shows a figure with multiple scales used, other figures show mixed mode data eg time series and frequency domin without sufficient (or legible labelling)
We agree that the X axis should normally be added to a figure. Therefore, we added the X axis to the signals shown in Figure 5. However, we believe that the X axis does not hold additional information for Figures 7, 9 and 10. Hence, we kept the figures as they were in the initial document and added an explanation in the captions of these figures on the time scale and on why the X axis is not shown.
- The review section is commendable however in the Table summaries a further column should be added that describes the methods used by the authors of such papers to improve its utility to the reader.
Thanks for the suggestion. An additional column was added that includes methods used in the listed papers.
- The table with the participate information would benefit through the inclusion of anthropomorphy as this is valuable to many researchers for analysis of data Examples here would be length length and arm length.
We agree that more concrete details on the players, such as arm and leg lengths, would be useful. However, the Ethics Council of our university explicitly demanded to not use more individuals' information apart from the descriptors that were given in the dataset description. We have added these limitations in Section 4, where the design of the study are described in detail.
- Section 5 describes various types of analysis however these tend to be solely black box numerical approaches, the paper would benefit by the inclusion of biomechanical derived methods. Where this is not possible stating reasons for exclusion and introducing using papers from the review section or something like
Thanks for the comment. We explained in the paper (line 685 - 689) why we did not include such analysis methods.
- Ethical clearance is required for the reporting of all data on human subjects. usually the approving institution and a clearance number is reported to ensure data is traceable to its origin and collection methods. Where historical data is used, even if ethics approval was not required by a commercial organisation, my understanding is that ethical clearance to access historical data is still required if it is to be reported in the literature.
An Institutional Review Board Statement is now added to the paper. The statement now includes the IRB reference number from both institutes in our resubmission.
Reviewer 3 Report
As a researcher in the field of HAR for many years, I sincerely congratulate you for your novel and substantial contribution to this aspect!
The points that need revision are:
The Informed Consent information, while indicated, should have had a more rigorous narrative regarding ethical approval, private information elimination or processing, pseudo-anonymity, etc., since the manuscript deals with human biosignals.
There are grammatical and expression flaws on almost every page, and I can even detect a lot of German English.
A few examples are
217 is BEING written.
221 vision-based (by the way, I don't understand why computer vision is capitalized here, if not to indicate the full name of the abbreviation)
Please double-check and proofread your manuscript carefully.
The literature part misses a recent CSL-SHARE dataset from Uni Bremen. This dataset is closely related and uniquely complementary to this manuscript's topic, for 1. It is the only HAR dataset that uses knee bandages as a carrier for wearables, which provides protection or rehabilitation for basketball players: the applied knee bandage from Bauerfeind, Germany, happens to be a care and rehabilitation partner contracted by the NBA; 2. This dataset, containing 20 participants' 22 daily + sports activities, involves several particularly relevant activities for basketball playing, such as left/right shuffling (e.g., in defense), four types of V-cuts (e.g., in layup), etc. These are not present in your dataset or any of Table 2's peer corpus. 3. The dataset also applied multiple IMUs, on the upper and lower limbs. Additionally, 4 EMGs, 1 goniometer, and even acoustic sensors were applied simultaneously, enriching HAR research's material; 4. On this strictly segmented and calibrated dataset, the accompanied work "how long are various types of daily activities" investigates a very informative topic for HAR, obtaining important results as a reference for, e.g., windowing and modeling.
See above
Author Response
Dear reviewer,
We first of all would like to thank you for taking the time to read our submission and provide us with your valuable expert opinions. In the past week, we have been working hard to address each of your reviews and specific feedback items. In the sections below, we answer how we have updated the submission according to your comments and questions, which accompany our resubmission where changes have been marked as well.
- The Informed Consent information, while indicated, should have had a more rigorous narrative regarding ethical approval, private information elimination or processing, pseudo-anonymity, etc., since the manuscript deals with human biosignals.
The Informed Consent has been updated with additional information about the content of the form signed by our participants.
- There are grammatical and expression flaws on almost every page.
The paper has now been proofread by a native English speaker.
- The literature part misses a recent CSL-SHARE dataset from Uni Bremen.
The dataset is now added to Table 1 and considered in the Discussion.
Round 2
Reviewer 1 Report
No additional comments
Reviewer 3 Report
The authors addressed my points well. I argue to accept this manuscript.